# Emotion Recognition for Partial Faces Using a Feature Vector Technique

**DOI:** 10.3390/s22124633

**Published:** 2022-06-19

**Authors:** Ratanak Khoeun, Ponlawat Chophuk, Krisana Chinnasarn

**Affiliations:** Faculty of Informatics, Burapha University, Chonburi 20131, Thailand; 60810065@go.buu.ac.th (R.K.); ponlawat.ch@go.buu.ac.th (P.C.)

**Keywords:** emotion recognition, landmark detector, facial mask, facial expression, occlusion, feature vectors

## Abstract

Wearing a facial mask is indispensable in the COVID-19 pandemic; however, it has tremendous effects on the performance of existing facial emotion recognition approaches. In this paper, we propose a feature vector technique comprising three main steps to recognize emotions from facial mask images. First, a synthetic mask is used to cover the facial input image. With only the upper part of the image showing, and including only the eyes, eyebrows, a portion of the bridge of the nose, and the forehead, the boundary and regional representation technique is applied. Second, a feature extraction technique based on our proposed rapid landmark detection method employing the infinity shape is utilized to flexibly extract a set of feature vectors that can effectively indicate the characteristics of the partially occluded masked face. Finally, those features, including the location of the detected landmarks and the Histograms of the Oriented Gradients, are brought into the classification process by adopting CNN and LSTM; the experimental results are then evaluated using images from the CK+ and RAF-DB data sets. As the result, our proposed method outperforms existing cutting-edge approaches and demonstrates better performance, achieving 99.30% and 95.58% accuracy on CK+ and RAF-DB, respectively.

## 1. Introduction

According to one study, masked faces are considered to be more reliable, causing people to reduce their interpersonal distance [1]. Moreover, emotion recognition and face perception are highly associated with one another, and challenges are even more severe for the people who have a hearing disability and who mostly communicate through facial expressions, especially through the mouth. Emotion recognition is important because it can be applied to a vast number of applications, such as those in [2,3,4,5,6,7], for problem solving and are of exceptional value to both organizations and communities. For instance, facial emotion recognition applications have been produced to improve human–computer interaction [8]. Those applications include stress detectors, driver fatigue detectors, attention assessments in online learning environments, physical pain detectors, and recommendation systems. Moreover, other scenarios such as criminal interrogation, driver fatigue detection, intelligence monitoring, telemedicine, and intelligent robots rely on the capability of emotion recognition [9].

In the context of emotion recognition from facial images, the most important feature representation approaches can be divided into two types [10]: face-based approaches that use face pixels as raw information, and constituent-based approaches that analyze the relationships between facial features.

For face-based approaches, the authors of [11] proposed multiple attention networks consisting of two sub-nets: the region-aware sub-net (RASnet) and the expression recognition sub-net (ERSnet), in order to locate critical regions and to learn discriminated features, respectively. Moreover, in order to enhance the discriminative power of deep features, the authors of [12] proposed the deep locality-preserving Convolutional Neural Network (DLP-CNN). On the other hand, a Convolutional Neural Network with an attention mechanism (ACNN) was proposed in [13]. The authors of [14] presented an end-to-end trainable Patch-Gated Convolution Neural Network (PG-CNN) to determine information from the occluded regions of the face while focusing on the discriminative and un-occluded regions. The authors of [15] proposed an Expression Net (eXnet) architecture as well as a Convolutional Neural Network (CNN) based on parallel feature extraction. The author of [16] adopted De-expression Residue Learning (DeRL) to extract expressive component information for the purposes of Facial Expression Recognition (FER). The approaches mentioned above have the ability to recognize facial emotions; however, they still essentially rely on the lower part of the face and are unable to detect and learn once the entire lower part of the face is completely covered by a facial mask. Furthermore, those methods still have high computational power and memory requirements, which are crucial for small hardware-constrained devices.

Conversely, for constituent-based approaches, Krishnaveni et al. [17] proposed a method for FER using low-level histogram features. Initially, the face region was detected using the Histograms of the Oriented Gradients (HOG) and a Support Vector Machine (SVM). HOG and LDP were applied in the feature extraction phase, and the features that were downsized by the soft-max layer were used to classify emotions. Allaert et al. [18] compared a variety of optical flow approaches. Cornejo et al. [8] proposed Robust Principal Component Analysis (RPCA) for facial expression reconstruction and adopted the Weber Local Descriptor (WLD), Local Binary Pattern (LBP), and Histograms of the Orientated Gradients (HOG) for the feature extraction process. Finally, emotion recognition was completed using the K-Nearest Neighbor (K-NN) and Support Vector Machine (SVM) approaches. Those approaches are straightforward. However, they only provide good accuracy for small data sets. Once a larger data set with a variety of constraints such an occlusion is used, the feature vectors of the approach will be full of outliner information, which is not only unnecessary information, but also produces a higher number of misclassified results, eventually leading to lower performance. Furthermore, optical flow techniques have the disadvantage of being highly sensitive in the presence of occlusions, light variations, or out-of-plane movements. For these reasons, occlusions such as a facial mask would pose even more challenges to these approaches.

In this paper, we proposed a rapid Landmark Detection approach for constituent-based facial recognition that is able to effectively describe the relationship between facial features and can recognize emotion in facial mask images. Our approach is straightforward, easy to implement, and has low computational complexity due to its ability to reduce the search space by a significant amount. A number of new features and the associations among them are discovered and will be explained in the following sections in detail. The rest of the paper comprises the following sections: Materials and Methods, Results and Discussion, and the Conclusion.

## 2. Materials and Methods

### 2.1. Analysis of Emotions in Facial Images

There are eight baseline emotions: neutral, contempt, happiness, fear, surprise, anger, disgust, and sadness, as shown in Figure 1. When the lower part of the face (nose, mouth, cheeks) is missing, “Surprise” may differ from the other emotions according to the shape and pattern of the eyes and eyebrows. However, “Anger” and “Disgust” are difficult to differentiate from one another. The upper parts of the face are similar, and without the lower part of the face, these two emotions can be misclassified. More importantly, distinguishing “Happiness”, “Fear”, and “Sadness” from one another can be rather agonizing because these three emotions crucially rely on the lower facial regions.

As shown in Figure 1b, each emotion has different characteristics according to the connections between the action units (AUs) of the facial image. When the lower part is not occluded, all of the facial features and the relationships between each and every landmark are clearly visible. However, when the lower part of the face is completely covered by a facial mask, only the upper part of the face is available for analysis. The challenge is that sadness, contempt, anger, disgust, and fear all appear very similar in the upper part of the face and are very hard to distinguish from one another: Contempt=∅, Neutral=∅, Surprise⊂Fear, Disgust⊂Fear, Anger⊂Fear, Sadness⊂Fear, Disgust⊂Anger.

A number of the existing approaches use a combination of specific AUs that include the upper and lower parts of the face to classify the emotions. However, as shown in Figure 1c, surprise, disgust, anger, and sadness are all proper subsets of fear, which is the reason why they can be misclassified. Moreover, contempt fully relies on the lower part of the face and will otherwise be misclassified as neutral. For this reason, the existing approaches that only rely on AUs are disrupted by the lack of information from the lower part of the face, which eventually leads to low accuracy. A comparison of the existing approaches is shown in Table 1.

Existing emotion recognition methods, such as those in [13,14,16,17], that use 68-landmark detection require the entire image to be searched in order to find the face and then label the locations of 68 landmarks on the frontal face. Those landmarks and their relationships are then used for analysis. However, those methods mainly rely on the relationship with the lower part of the face; therefore, achievement is disrupted once the lower part is occluded, which means that only approximately 38% of the information is available for those methods.

For those face-based methods, all of the pixels in the detected faces are used to learn and classify emotions, resulting in high complexity and computational time. The problems with face-based methods are that all of the pixels are used, and those unrelated pixels are not relevant to the process. Moreover, they also disrupt the machine learning process and consequently lead to low accuracy and high complexity.

As shown in Figure 2, different emotional landmarks are detected separately. Due to the fact that each and every CK+ participant performed fewer than the eight total emotions, the participant in Figure 2 is shown presenting seven of the emotions, excluding contempt. For the existing approaches that use the entire frontal facial landmarks to classify emotions using Equation (2), there are a number of problems. Due to the fixed number of landmarks, the same number of landmarks for the different emotions of a single person is produced. That being said, what makes each emotion distinct from one another is not about the number of landmarks and is instead about the location and other parameters at each landmark.

As shown in Figure 3, a number of samples were randomly selected from CK+ to analyze the density and distribution of the landmarks in each sample. We observed that both the lower and upper parts of the histograms of distance for the landmarks across all of the emotions observed in Samples 1–6 are right-skewed distributions. On the other hand, Samples 7–9 are not right-skewed; however, the distances between the lower parts of Samples 1–9 are significantly higher than those of the upper parts. More importantly, the histograms of distance between the landmarks across all of emotions in the CK+ samples depicted in Figure 4a show that the distances between the landmarks in the lower part of the face are significantly higher than those for the upper part of the face between emotions. Additionally, the distances of all of the landmarks across all emotions were sorted and are shown in Figure 4b. The figure shows that the distances between the landmarks for the lower part of the face are higher than those for the upper parts.

Once all of the landmarks of a person’s emotions are combined together, the lower part of the face either demonstrates high density or high distribution among the landmarks. To be more precise, by only using the landmarks from the lower part of the frontal face, emotions can be easily classified due to the fact that the locations of the landmarks differ significantly from one another within different emotions. However, the characteristics of the landmarks in the lower part do not exist in the upper part of the face, which means that in the upper face, those landmarks either have a low density or distribution. This causes the location of the upper facial landmarks be similar to the landmarks labeled for different emotions, as seen in Equation (1), which determines the Euclidean distance. As such, with a fixed number of landmarks and low density and distribution, it is difficult to distinguish between emotions using the upper part of the face alone. Moreover, this is the only scenario where different emotions from the same person are compared, which means that the problem of distinguishing emotions from different people is even more challenging.
(1)$$d\left(L_{m,Emotion1},L_{m,Emotion2}\right)=\sqrt{{\left(x_{Emotion2}-x_{Emotion1}\right)}^{2}+{\left(y_{Emotion2}-y_{Emotion1}\right)}^{2}}$$
(2)$$Landmark_{Traditional}=Upper\;\cup \;Lower$$
(3)$$Upper=\left\{L_{1},L_{17}\right\}\;\cup \;\left\{L_{18},L_{19},\ldots ,L_{22}\right\}\cup \;\left\{L_{23},L_{24},\ldots ,L_{27}\right\}\;\cup \;\left\{L_{28},L_{29}\right\}\;\cup \;\left\{L_{37},L_{38},\ldots ,L_{48}\right\}$$
(4)$$Lower=\left\{L_{2},L_{3},\ldots ,\;L_{16}\right\}\;\cup \;\left\{L_{30},L_{31},\ldots ,L_{36}\right\}\cup \;\left\{L_{49},L_{50},\ldots ,L_{68}\right\}$$
(5)$$\forall L_{m}\in Upper\Rightarrow d\left(L_{m,Emotion1},L_{m,Emotion2}\right)\to 0,\;\;where\;m\in \left\{1,\;2,\;\ldots ,\;68\right\}$$
(6)$$\forall L_{m}\in Lower\Rightarrow d\left(L_{m,Emotion1},L_{m,Emotion2}\right)\to \varphi ,\;\;where\;m\in \left\{1,\;2,\;\ldots ,\;68\right\},\;\varphi \gg 0$$

As shown in Figure 5, in our scenario, the entire lower part of the face is completely occluded by a facial mask; this is represented by Equation (4). More importantly, in the lower part of the face where the landmarks have a higher density and high distribution, as represented by Equation (6), only 26 landmarks or 38% still occur, and 42 landmarks of the original 68 landmarks in Equation (4) have completely vanished. As depicted in the combined upper facial landmarks for all of the emotions, the landmarks on the eyebrows and nose, $\left\{L_{1},L_{17}\right\}\;\cup \;\left\{L_{18},L_{19},\ldots ,L_{22}\right\}\cup \;\left\{L_{23},L_{24},\ldots ,L_{27}\right\}\cup \;\left\{L_{28},L_{29}\right\},$ as per Equation (3), for all emotions are all close to one another. This means that the distances between the same labeled landmarks are significantly smaller for both the *x* and *y* coordinates, as shown in Equation (5). Only 12 landmarks in the eye area, $\left\{L_{37},L_{38},\ldots ,L_{48}\right\},$ or 18% of the total traditional landmarks, are slightly different between emotions, which means that very little significant information is left for emotion recognition.

From the above-mentioned problems resulting from traditional landmarks, in order to eliminate the problems where the number of landmarks produce insignificant characteristics, we propose an approach that has the ability to flexibly detect a number of significant upper facial landmarks without a fixed number and location of landmarks, as shown in Figure 6. As per Equation (7), our detected landmarks do not have to be located around the eyebrows, eyes, or nose. Our process does not require us to take all the pixels from the upper face. On the other hand, the upper face happens to be where our generated paths travel through. Without the fixed number of landmarks, the detected number of landmarks representing the different emotions of a single person or even for multiple people can be used as a significant feature in the emotion recognition process. More importantly, rather than focusing on the information around the fixed landmark area, enormous amounts of information from the area of the flexible landmarks that have been previously detected can be studied.
(7)Our Dectected Upper Landmark={L1,…,Ln}, where n∈ℕ

Our approach is flexible, has low complexity as well as simple implementation, and most importantly, it can achieve high accuracy. We would like to determine the important patterns that exist across the upper part of the face that are available when wearing a mask. We thus propose an idea in which the generated path can detect the intensity pattern in a flexible way, reducing the complexity of the existing emotion recognition methods that use all of the pixels from both the upper and lower parts of the face. The generated path creates a shape that looks similar to an infinity shape, and which needs to travel across the eye and eyebrow regions multiple times in slightly different neighborhoods for different periods. For this reason, a single round is not enough to gather essential information. Paths need to be travelled multiple times. We propose a set of equations, Equations (8) and (9), and properties in order achieve this. A small set of points generated from the equations runs across the eye and eyebrow areas, detecting significant amounts of information as well as patterns along the way; the generated points are then able to successfully recognize the emotions with a high degree of accuracy and low complexity.
(8)x=cos(t),
(9)y=sin(αt)β
where 0≤ti≤cπ , 0≤i<n , and i∈ℕ; α is the number of circular-like regions produced by the equations, β defines the height of the shape, and n is the number of generated points to create the shape.

As shown in Figure 7, Equations (8) and (9) are plotted separately. The blue and red graphs represent the graphs of cos(t) and (sin(2t))/2, respectively. We observe that in order to complete a single period (T1) of cos(t), it takes the graph of (sin(2t))/2 two complete periods (T2) to achieve the same length. In this scenario, once we combine Equations (8) and (9) together and define x=cos(t) and y= sin(2t)/2, we obtain a graph with an infinity shape, as shown in Figure 7. The infinity shape shows that the shape requires 50 points to complete a single period (T1) of x= cos(t), in which $0\leq t_{i}\leq 2\pi$, and two periods (T2) of y= sin(2t)/2, which means that the graph needs to travel through coordinate x five times and through coordinate *y* two times.

From the above-mentioned scenarios, our purpose is to generate multiple periods (rounds) in which the points of each round land on a different path from one another in order to create an ideal area that covers a number of significant regions of the upper face in the next step. As such, the idea is to increase the number of periods so that the graph can travel over multiple rounds. However, if the number of points (*n*) increases to have the same ratio as the number of periods (*c*), the result will be a look-alike single graph. From our experiments, the ideal setting for our infinity shape requires that the number of periods be increased to *c* = 20, which makes $0\leq t_{i}\leq 10\times 2\pi ;$ this means that the graph depicts 10 periods (rounds). Moreover, the number of points needs to be stable; in this case, *n* = 50. Hence, there are five points that land on a completely different path from the others in each round.

### 2.2. Proposed Method

When it comes to recognizing emotions without being able to see the lower part of the face, significant information can only be extracted from the upper part of the face. We focused on the eye and eyebrow regions since the upper part of the face receives the information spreading from the lower-face region. We observed that both of the eyes form a shape that looks similar to an infinity symbol (∞), as shown in Figure 7. As such, even if we do not have the lower facial features available, with the right technique, the upper area of the face alone can provide enough information for emotion recognition.

The illustration of the proposed method is shown in Figure 8, and our proposed feature extraction algorithm is shown in Algorithm 1. Moreover, each step will be described in the following sections in detail.
**Algorithm 1:** Landmark Detection and Feature Extraction**Input:** Upper facial image**Output:** Landmark coordinate and HOG feature landmarkst ← [0, 20π50 , 2×20π50 , 3×20π50 ,…, 20π] x ← cost y ← sin4t2 points ← (x, y)**for** i=0 to (length of t) − 3 **do**   startPoint ← points[i]   midPoint ← points[i+1]   endPoint ← points[i+2]   connectedLineIntensity[i] ← connect(startPoint, midPoint, endPoint)**end for** height ← height of peakfwhm ← full width of half maximum**for** each lineIntensity in connectedLineIntensity **do**   (indices, properties) ← findPeaks(lineIntensity)
**end for****for** i=0 to length of properties **do**   **if** prominence of properties[i] >= height    **and** width of properties[i]>=fwhm **then**     property ← properties[i]     landmarks[i] ← property[leftBase]   **end****end for****for** i=0 to length of landmarks **do**  landmarkHOG[i] ← histogramofOrientedGradient(landmarks[i])  features[i] ← (landmark[i], landmarkHOG[i])**end for**

#### 2.2.1. Synthetic Masked Face and Upper Face Detection

In this step, we wanted to generate a facial image with a synthetic mask using the original facial image from the CK+ [29] and RAF-DB [30] data sets. The Dlib face detector [31] was employed to detect the frontal face area from the entire image. We then obtained the different-sized facial regions according to the facial size of each image. After that, the facial mask was placed over the lower region of the detected face, including the nose, mouth, and cheeks, to generate a synthetically masked facial image that was as similar to a real-life scenario as possible, as shown in Figure 9. Finally, only the upper part of the masked facial image was detected and used in the next steps. For the CK+ data set, the detected upper parts varied in dimension. However, the average dimensions were approximately 236 × 96. For the RAF-DB dataset, the dimensions of the detected upper part were 100 × 46.

#### 2.2.2. Generated Points Creating Infinity Shape

We wanted to develop a rapid facial landmark detector that was free from issues related to occluded lower facial features. From our observations, the occluded parts (mouth, cheek, and nose area) of the face spread naturally in relation to the unobstructed parts (eyes and eyebrows area) when an emotion is expressed. We also wanted to develop a new landmark detector and extract the facial feature vectors that represent the important relationships between those regions in order to classify emotions. Both of the eyes form a shape that resembles an infinity symbol (∞). A combination of the simple yet effective trigonometric equations, such as those in Equations (8) and (9), were adopted to fulfill our objectives. In this step, we wanted to make sure that the generated points covered the area outside of the eyes and eyebrows. The reason for this is that the important features that occur between the lines connecting neighborhood points need to be discovered instead of the entire pixel area being used to train and classify different emotions. By doing this, the computational complexity is significantly reduced.

For our infinity shape, the number of periods needs to be α = 2, β = 2, and c = 20, which makes $0\leq t_{i}\leq 10\times 2\pi$, thus creating 10 graphing periods (rounds). Once β = 2, two circular regions are generated. Moreover, the number of points needs to be stable at *n* = 50 so that there are five points in each round landing on a completely different path from the others, as shown in Figure 7.

#### 2.2.3. Infinity Shape Normalization

The original generated set of points of the infinity shape are different in size from the upper face image. In this step, we need to normalize the size of the original infinity shape so that it is the appropriate size before moving it to the location where it can essentially cover the upper face region, including the eyes and eyebrows.

As shown in Equations (10) and (11), in order to calculate the new coordinates of each and every point, we take the original coordinate value of either *x* or *y* and transform it into a different range or size according to each individual upper face area. In this case, the new range $\left(\max_{new}-\min_{new}\right)$ is actually the width or height of the upper face. By doing this, we can obtain the normalized *x* and *y* coordinates, in which the size of the infinity shape is in the same range as the upper face area. This is also one of the reasons why our proposed method is very flexible: the original set of points is normalized as the sizes of different upper face regions independently.
(10)$$x_{normalized}=\frac{x_{ori}-\min_{x_{ori}}}{\max_{x_{ori}}-\min_{x_{ori}}}\left(\max_{x\_new}-\min_{x\_new}\right)+\min_{x\_new}$$
(11)$$y_{normalized}=\frac{y_{ori}-\min_{y\_ori}}{\max_{y\_ori}-\min_{y\_ori}}\left(\max_{y_{new}}-\min_{y\_new}\right)+\min_{y\_new}$$

#### 2.2.4. Initial Seed Point of Upper Face

At this point, our challenge was to find the initial seed point and to move the entire normalized infinity shape to its new position. By doing this, the connected lines would cover the important areas along the way, including the eyes and eyebrows. In finding the initial seed point, we discovered that this point is where all of the forces created by the connected lines gather together. Because all of the vectors point to the same direction, the initial seed point has a huge number of accumulated forces from the infinity shape.

On the other hand, when we considered vertical and horizontal projections, we discovered that they actually lead to the same conclusion regarding where the initial seed point is. The vertical projection is the summation of all of the intensity values from each column, while the horizontal projection is the summation of all of the intensity values from each row. The reason for this is that we observed that the nose area is significantly brighter than the other areas in the vertical scenario. On the other hand, the eye and eyebrow areas have a lower intensity compared to the other areas in the horizontal scenario.
(12)$$x_{seed}=index\left(vertical_{\max}\right)$$
(13)$$y_{seed}=index\left(horizontal_{\min}\right)\;$$

We needed to find the coordinates of the seed point $\left(x_{seed},\;y_{seed}\right)$, as per Equations (12) and (13). This worked well for the $x_{seed}$ because the nose area is always the brightest area compared to the other areas. However, there are three different cases that can be observed from the experiment in the horizontal scenario: When the eye area is darker than the eyebrow, the minimum value of the horizontal projection lands on the eye area. Additionally, it lands on the eyebrow area when the eyebrow area is darker than the eye area. The last and best case happens when the eyes and eyebrows are blended into one region, and when the minimum intensity value lands on the area between the eyes and the eyebrows. In order to overcome this problem, we smoothed the horizontal projection graph using the Savitzky–Golay filter [32], with the window size set to 31 and the order of the polynomial used during filtering set to 3. By smoothing the horizontal projection graph, the three cases are generalized into a single case, in which Equations (12) and (13) can be applied; we can then successfully determine the coordinates of the seed point $\left(x_{seed},\;y_{seed}\right)$.

#### 2.2.5. Landmark Detection (Eye and Eyebrow Boundary Detection)

In this step, once the generated points have been normalized and moved, the correct seed point automatically covers the edges of the eyes and eyebrows, and the intensity values along all of the connected lines are then extracted to be able to analyze the characteristics of each line, as shown in Figure 10.

Once the intensity values had all been extracted and concatenated, we then observed that there was a certain pattern that could be observed inside the graph. We determined that patterns with a high prominence value (L1) as well as with a high-prominence width (L2) correspond to our original assumption perfectly, since the high prominence value (L1) means that the intensity is changing from low to high to low, or that the peak value is significantly high, which means that two connected lines of our infinity shape are travelling over a significant edge. A high prominence value alone is not enough, and the wide prominence width (L2) is also taken into consideration. A wide prominence width indicates the presence of a large number of pixels travelling across a significant edge.

We then focused on all of the three-neighborhood points and generated two subsequence graphs of points $P_{t-1}$ to $P_{t}$ and points $P_{t}$ to $P_{t+1}$. After the graph had been generated, the search process was adopted in order to search for the pattern. The detected and undetected landmark patterns are shown in Figure 10. Once the pattern was detected from each connected graph, the index of point A was then considered as the point where the intensity changes from low to high. This means that A is the point of the edge or the boundary of the eye area. We then plotted all of those detected points to the image, and they landed perfectly around the eyes and other significant areas, as shown in Figure 10.

#### 2.2.6. Feature Extraction (Histograms of the Oriented Gradients)

As shown in Figure 11, most of the detected upper face landmarks have coordinates that correspond to the boundaries of the eye and eyebrow areas. Besides the coordinates of each and every landmark, we believe that the relationship between those detected landmarks are also the significant features for the classification process. The detected points are considered to be candidate landmarks as there are still a few outliers. The purpose of this step is not only to perform the feature extraction process, but also to eliminate the outliners. As such, we decided to determine the relationship between the candidate landmarks by applying the Histograms of the Oriented Gradients (HOG), as per Equations (14)–(17), to each and every upper face landmark in order to obtain the direction and magnitude value of each direction, as shown in Algorithm 2. The data obtained from each landmark are the coordinates of the landmark (x, y), followed by the 72 values of the direction and the magnitude of the 20 × 20 blob size around the landmark. This means that the 74-feature vector can be detected from each landmark. By doing this, the effective information extracted from each landmark will contain information on the relationship that exists between those neighborhood landmarks.
**Algorithm 2:** Histogram of Oriented Gradient (HOG)**Input:** Upper face image, boundary coordinate, size of blob**Output:** HOG feature of each boundary coordinateimage ← Upper face imagex_landmark ← x coordinate of boundaryy_landmark ← y coordinate of boundarydistance ← size_of_blob/2**for** i=0 to length of boundary coordinate **do**  **if** x_landmark[i] > distance **then**    x_start_blob ← x_landmark[i] − distance  **else**    x_start_blob ← 0  **end**  **if** y_landmark[i] > distance **then**    y_start_blob ← y_landmark[i] − distance  **else**    y_start_blob ← 0  **end**  x_end_blob ← x_start_blob + size_of_blob  y_end_blob ← y_start_blob + size_of_blob  blob ← image[x_start_blob, x_end_blob][y_start, y_end_blob]  hog_feature[i] ← HOG(blob)**end for**

The HOG is one of the quality descriptors that aims to generalize the features of any of the objects, in which the same objects produce the same characteristic descriptor in different scenarios. The HOG works with the numbers in the local gradient directions of time that have a large magnitude value. It utilizes the gradient of an image, the set of histogram directions of a number of locations, and the normalization of the local histograms. Furthermore, the local histograms need to be localized to the block. Without any clues to elucidate the edge positions or the equivalent gradient, the key idea is that the allocation of the border direction (local intensity gradients) characterizes local object manifestation and form, as shown in Figure 11.
(14)$$G_{x}\left(y,x\right)=Y\left(y,x+1\right)-Y\left(y,x-1\right)$$
(15)$$G_{y}\left(y,x\right)=Y\left(y+1,x\right)-Y\left(y-1,x\right)$$
(16)$$G\left(y,x\right)=\sqrt{G_{x}{\left(y,x\right)}^{2}+G_{y}{\left(y,x\right)}^{2}}$$
(17)$$\theta \left(y,x\right)=\arctan\left[\frac{G_{y}\left(y,x\right)}{G_{x}\left(y,x\right)}\right]$$

The detected HOG features of each landmark are concatenated into a single vector. The combination of the coordinates (x, y) of each landmark and the HOG feature vector is considered to be the representative data for each landmark. All of the landmark data from every image with different emotions are then labeled before entering the classification step.

#### 2.2.7. Classification

For the classification step, the features from each landmark are combined together and entered into the training process. As shown in Figure 12b, we adopted CNN, LSTM, and ANN to train and classify the data. As each facial image has a number of landmarks and each landmark has multiple features, the data are considered in one training window. When it comes to multiple training windows, LSTM is adopted. In this case, Long Short-Term Memory (LSTM) is applied after flattening the layer of our architecture in order to determine the relationship between those HOG features and each detected landmark, since LSTM has the ability to remember the knowledge obtained from the previous sliding window.

In Figure 12a, the σ are the gates of the LSTM components and are the neural networks that indicate the amount of information obtained from the predictors and the previous cell. These gates pass the information to the tanh functions to update the weight of the nodes. After that, the long-term memory (cell state) and the short-term memory (hidden state) are produced. To be more precise, the forget gate decides how much long-term information should be forgotten. The information that is kept is the input to the cell state. Moreover, for the input gate, it specifies how much short-term information should be forgotten. Then, the information is pushed through to the tanh functions for training. Finally, the output gate uses both the short and long-term information to determine how much information to transfer into the hidden state, as per Equation (18).
(18)$$c_{t}=f_{t}⊙c_{t-1}+i_{t}⊙{\tilde{c}}_{t}$$
where $f_{t}$ is the forget gate, $c_{t-1}$ is the prior cell state, $i_{t}$ is the input gate, and ${\tilde{c}}_{t}$ is the intermediary cell state.
(19)$$f_{t}={\left(\frac{t-k_{t}+1}{t-k_{t}}\right)}^{-p}$$

The forget gate in Equation (19) determines how far back to look in the long-term memory, as t is the index of the current timestep, $k_{t}$ is the index of the first timestamp of interest, and p controls the rate of information decay.
(20)$$k_{t}=r_{t}⊙t+(1-r_{t})⊙k_{t-1}$$

As shown in Equation (20), $r_{t}$ is the reset gate, which is a sigmoid activation function. To be more precise, this means that the gate takes a value between 0 and 1. In the case when the value is 1, $k_{t-1}$ is canceled out, and all prior information is removed. If its value is 0, t is canceled out, and the historical information is kept.

## 3. Results and Discussions

In our experiment, the CK+ [29] and RAF-DB [30] data sets were used. These two popular data sets were used because they vary in terms of the age, gender, nationality, head posture, lighting conditions, and occlusion in the images. For instance, the CK+ data set contains participants from 18 to 50 years of age from a number of nations, and 69% of the images are of women, presenting a great variety and posing a challenge for us. The CK+ data set only contains frontal facial images, which provided us with the possibility to study normal faces in normal environments. On the other hand, the RAF-DB data set is also one of the largest facial expression databases in the wild that contains a great number of head postures, lighting conditions, and occlusions. It provided us with the challenge of studying real-life facial images that are full of obstacles. Thus, the combination of the CK+ and RAF-DB data sets was ideal for covering a large number of real-life scenarios.

Moreover, unlike other approaches in Table 2, where both the lower and upper parts of the non-masked face were used as the training input, we extracted features of our proposed infinity shape only from the upper part of the synthetic masked face image. As shown in Figure 13, the training and validation learning curves of loss and accuracy that were determined using data sets from both databases show that learning data have a good fit with the scenario, with the training and validation losses stabilizing at a point where the difference between the two final loss values is low. To be more precise, the landmarks detected from our infinity shape and the HOG features obtained from each one of them are effectively significant and meaningful for emotion classification. As for the overall results, we achieved 99.30% and 95.58% accuracy for the data set from the CK+ and RAF-DB databases, respectively, as shown in Table 2. The accuracy metrics of the proposed method for CK+ and RAF-DB are shown in Table 3. Moreover, the comparison of the accuracy metrics of the proposed method used on the CK+ and RAF-DB data sets is shown in Figure 14 and includes precision, recall, f1-score, and accuracy. However, since contempt is not available in the RAF-DB data set, only the accuracy metrics for CK+ are shown in the last graph of Figure 14.

The CK+ [29] data set was used in our work. It consists of 593 acted facial expression sequences from 123 participants and covers eight baseline emotions: neutral, anger, contempt, disgust, fear, happiness, sadness, and surprise. The participants from the CK+ data set were 18–50 years of age, 69% female, 81% Euro-American, 13% Afro-American, and 6% were from other groups. The dimensions of the images were 640 × 490. The expression sequences begin in a neutral state and end in an apex state. However, our research focused on recognizing emotions using a single image, so only the apex image of each participant was used to perform our approach. Moreover, some of the participants performed more than one emotion, which means that more than a one image could be obtained from those participants. On the other hand, the Real-World Affective Faces Database (RAF-DB) [30] is one of the largest facial expression databases in the wild. The RAF-DB contains approximately 30,000 facial images. The images were collected from the internet. Each image was independently annotated with basic or compound expressions by 40 annotators. Only the basic expression images containing the seven baseline emotions: neutral, anger, disgust, fear, happiness, sadness, and surprise, were used in this research. The dimensions of the aligned images were 100 × 100. There is wide variation in the images in terms of age, gender and ethnicity, head posture, lighting conditions, occlusion (e.g., glasses, facial hair, or self-occlusion), and post-processing operations (e.g., various filters and effects). The participants in this data set were Caucasian, African American, and Asian. Five age ranges were included in the data set: 0–3, 4–19, 20–39, 40–69, and 70+ years of age.

The results show that the combination of the locations of the landmarks and the HOG features of the landmarks are significant assets that can be used to classify emotions because the HOG calculates both the horizontal and vertical components of the gradient magnitude and the direction of each individual pixel. According to [33,34,35], the HOG can effectively extract facial features. Unlike existing approaches where the HOG is calculated using the entire face, only the area around the detected landmarks defined by our proposed infinity shape were used to calculate the HOG. Due to the fact that the proposed detected landmarks are independently located, close to one another, as shown in Figure 15, and have a 20 × 20 blob size area around the landmarks used to calculate the HOG, a number of overlapped areas were used for the HOG at different landmarks. For this reason, similar HOG characteristics occurred within the neighborhood of each detected landmark.

Moreover, the proposed method is able to adapt to the dimensions of the face due to the fact that the proposed infinity shape is automatically moved to the initial seed point and scaled to the adaptive size according to the size of the upper part of each individual image. To be more precise, once the infinity shape has been adapted to the dimensions of the face, it will automatically cover the important parts of the face that contain the pattern of the prominence value and width and provide the same number of landmarks and their location for each sample, which is essential for recognition accuracy.

Therefore, our proposed set of points in an infinity shape surprisingly outperforms other existing methods in terms of its ability to detect the landmark locations that are full of significant information collected from the upper face regions alone. Our proposed method can flexibly detect the landmarks in the areas along the path that the infinity shape follows when traveling over the upper face area, especially the eyes and eyebrows. Once the path travels through the significant intensity patterns, certain spots are defined as landmarks, even when entire pixels of the upper face have not been computed. This means that the amount of data is reduced and that the method has low complexity and a low computational time.

For some scenarios, such as surprise, where the eyebrows are raised and are not covered by our infinity shape, our approach has the ability to detect other significant landmarks within the eye area or the area between the eyes that are available and can bring the landmarks into the classification process. Moreover, for some other scenarios, such as contempt or sadness, where the eyebrows are lower than the infinity shape, the infinity shape does not cover the eyebrows; however, the path still travels through the eye area, and the landmarks can be detected. However, in scenarios in which the eye area or the upper part of the face is too dark, or when the contrast is very low, the pattern cannot be detected by our infinity shape, which is an issue that needs to be fixed in our future work.

## 4. Conclusions

A large number of image processing and computer vision approaches, especially those for emotion recognition, have been distracted by a new global challenge in the recent years. People have been directed to wear a facial mask in public places. This new challenge excludes the presence of the most effective facial features (mouth, cheeks, and nose) and decreases the performance accuracy of existing emotion recognition approaches. In this paper, in order to recognize emotion in images with facial masks, we proposed a rapid feature vector technique consisting of three main steps. First, we took the input image and applied the preprocessing step to add a synthetic mask that covered the lower part of the face, and only the upper part of the facial image was taken into consideration. Secondly, the feature extraction technique based on our proposed rapid landmark detection technique that employed an infinity shape was utilized to flexibly extract a set of feature vectors that effectively indicated the characteristics of the partially occluded masked face. Finally, those features, including the location of the detected landmarks and the Histograms of the Oriented Gradients, were introduced into the classification process by adopting CNN and LSTM, and the experimental results were then evaluated using images from the CK+ [29] and RAF-DB [30] data sets. With our proposed landmark detection technique, our proposed method outperformed existing cutting-edge approaches and provided significantly higher performance in many aspects, achieving 99.30% and 95.58% accuracy for the CK+ and RAF-DB data sets, respectively.

More importantly, our future work aims not only to explore more rapid vectors as well as feature vectors obtained from a single facial image, but also image sequencing (video), which presents indispensable characteristics of occluded faces for facial recognition and other applications.

## Figures and Tables

**Figure 1 sensors-22-04633-f001:**
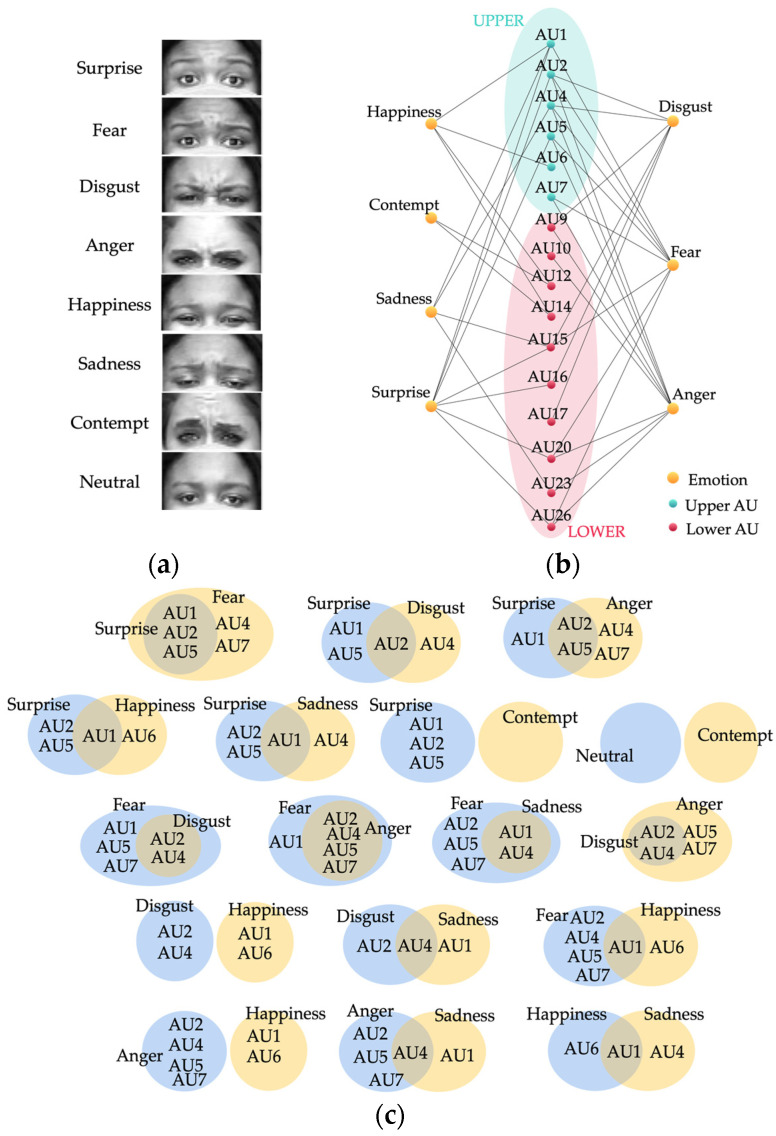
Upper portion of the face and AU samples of the baseline emotions; (**a**) upper face; (**b**) AUs of each emotion; (**c**) comparison of the upper AUs of each pair of emotions.

**Figure 2 sensors-22-04633-f002:**
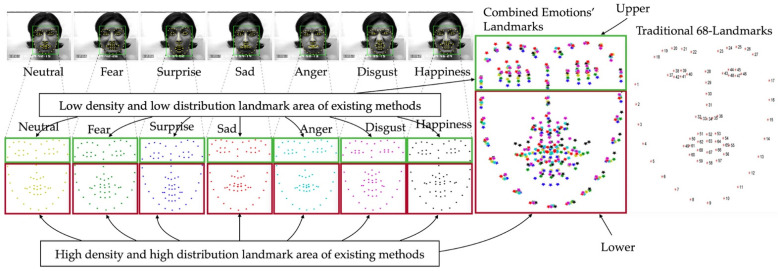
Traditional landmarks.

**Figure 3 sensors-22-04633-f003:**
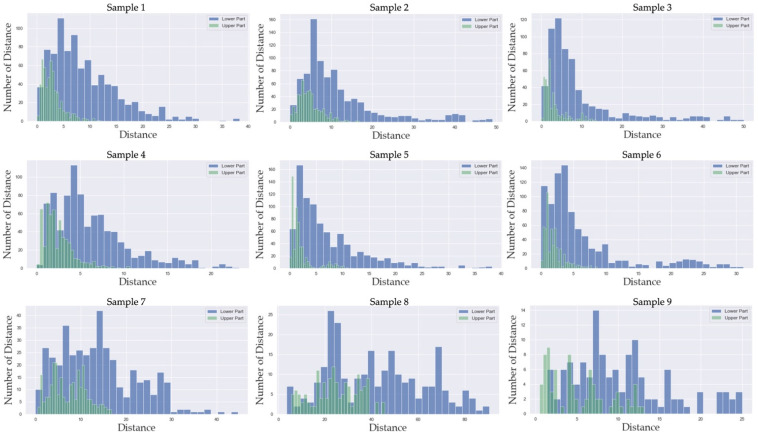
Histograms of the distance between landmarks across all emotions from nine CK+ samples.

**Figure 4 sensors-22-04633-f004:**
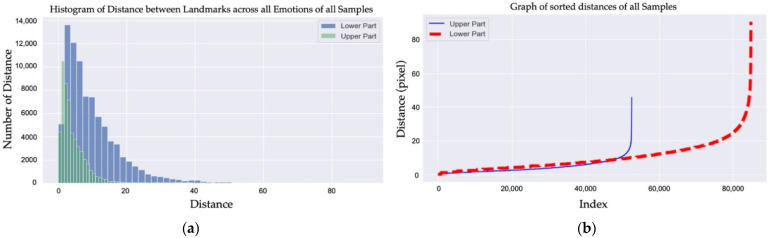
(**a**) Histograms of the distance between landmarks across all emotions from all of the samples in CK+; (**b**) graph of the sorted distances of all of the samples in CK+.

**Figure 5 sensors-22-04633-f005:**
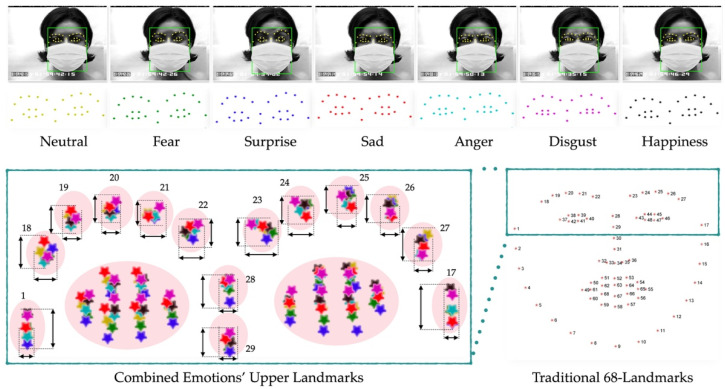
Each traditional upper landmark has a similar location to one another across the different emotions of a person; $Neutral_{1,17,18,\ldots ,29}≅Fear_{1,17,18,\ldots ,29}≅Surprise_{1,17,18,\ldots ,29}≅Sad_{1,17,18,\ldots ,29}≅Anger_{1,17,18,\ldots ,29}≅Disgust_{1,17,18,\ldots ,29}≅Happiness_{1,17,18,\ldots ,29}≅Contempt_{1,17,18,\ldots ,29}$.

**Figure 6 sensors-22-04633-f006:**
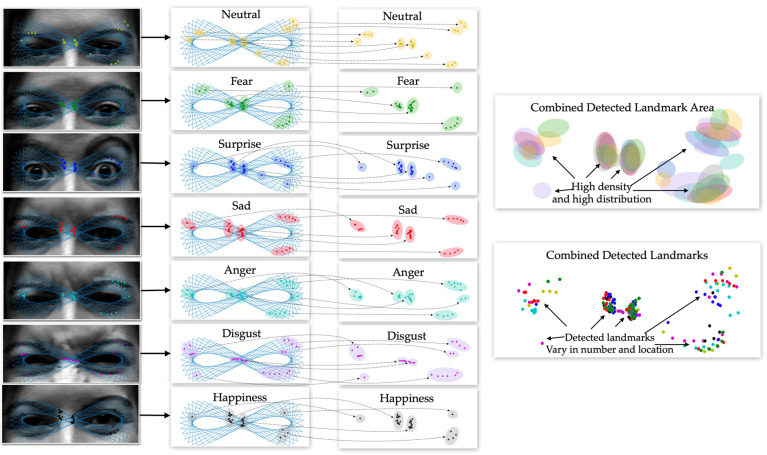
Landmarks detected using our proposed method.

**Figure 7 sensors-22-04633-f007:**
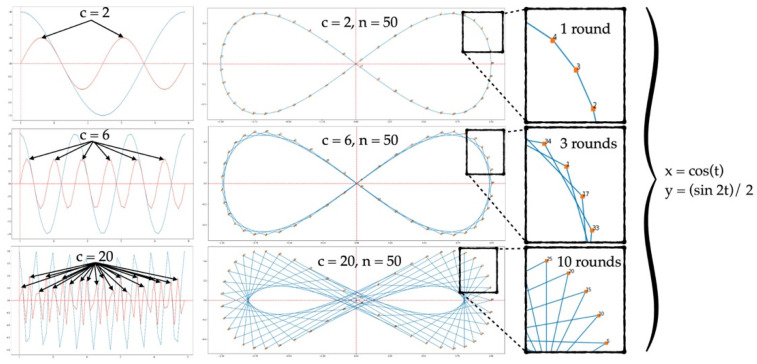
Illustration of infinity shapes.

**Figure 8 sensors-22-04633-f008:**
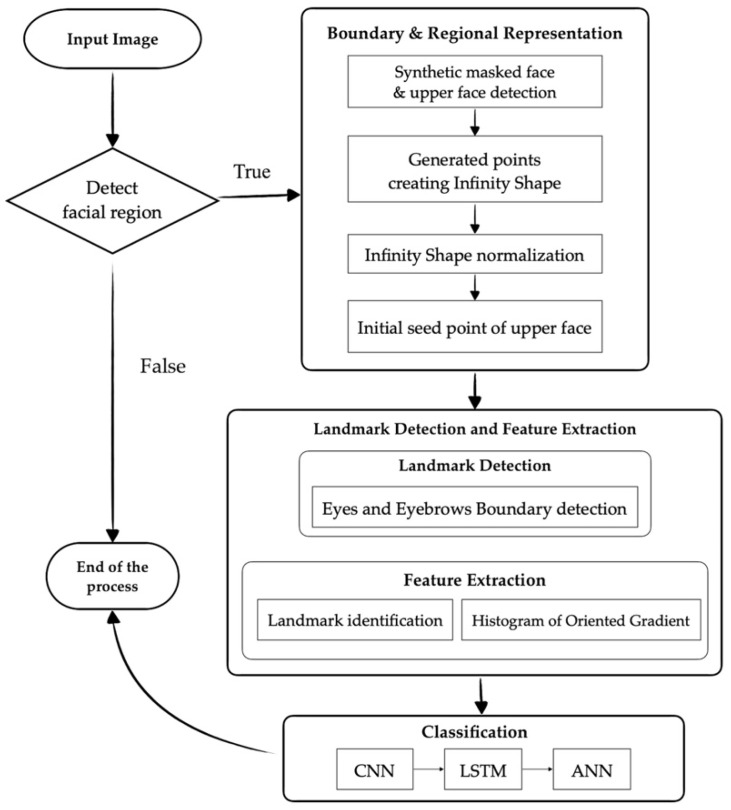
Illustration of the proposed method and the algorithm.

**Figure 9 sensors-22-04633-f009:**
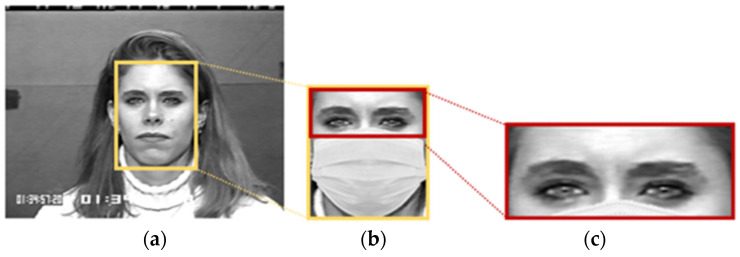
Upper face detection; (**a**) CK+ original image; (**b**) face detected using Dlib with the overlapping synthetic facial mask; (**c**) detected upper face.

**Figure 10 sensors-22-04633-f010:**
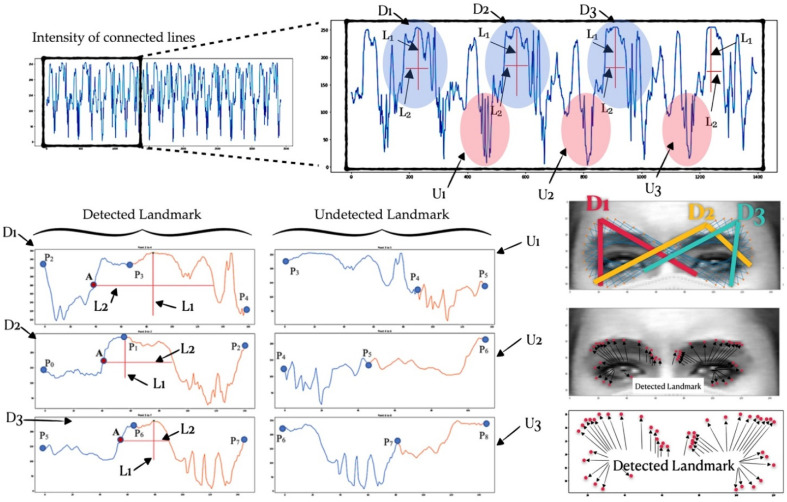
Detected and undetected landmarks.

**Figure 11 sensors-22-04633-f011:**
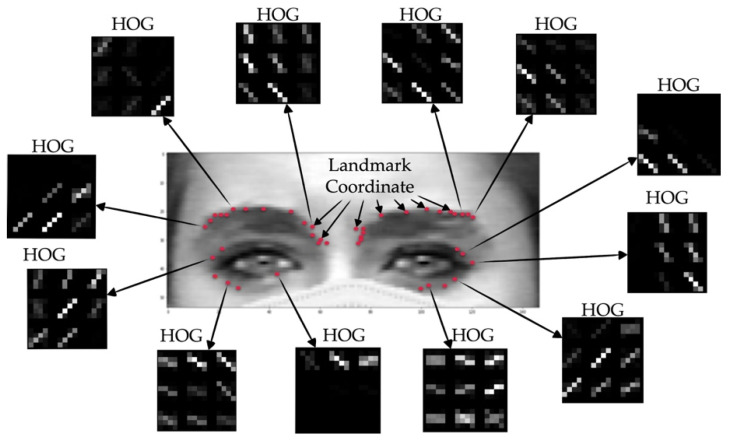
Example of the HOG features of each detected landmark, the HOG feature vector, and the algorithm.

**Figure 12 sensors-22-04633-f012:**
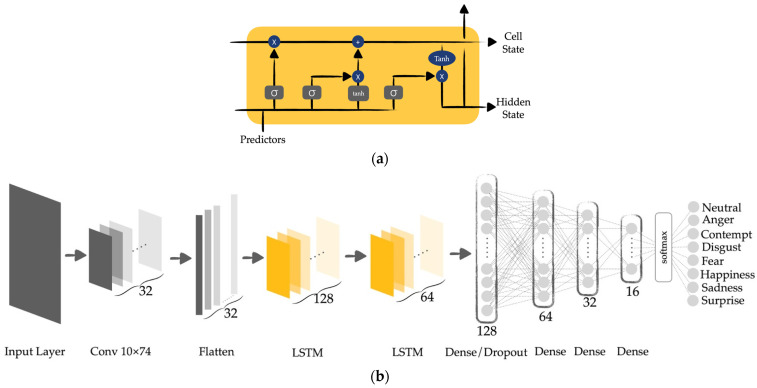
(**a**) LSTM components; (**b**) machine learning architecture.

**Figure 13 sensors-22-04633-f013:**
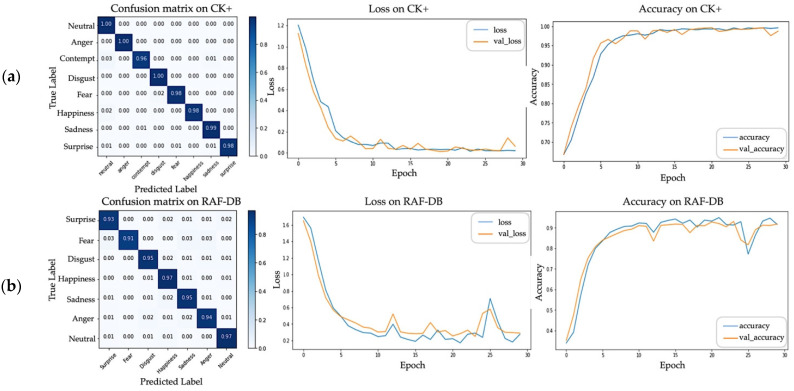
Results of our proposed method, including the confusion matrix as well as the loss and accuracy graphs; (**a**) results for CK+; (**b**) results for RAF-DB.

**Figure 14 sensors-22-04633-f014:**
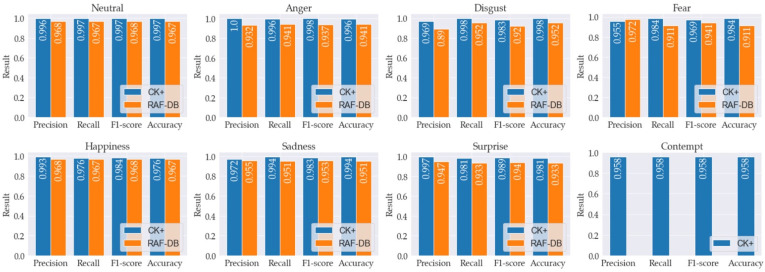
Comparison of the accuracy metrics of the proposed method used on the CK+ and RAF-DB data sets.

**Figure 15 sensors-22-04633-f015:**
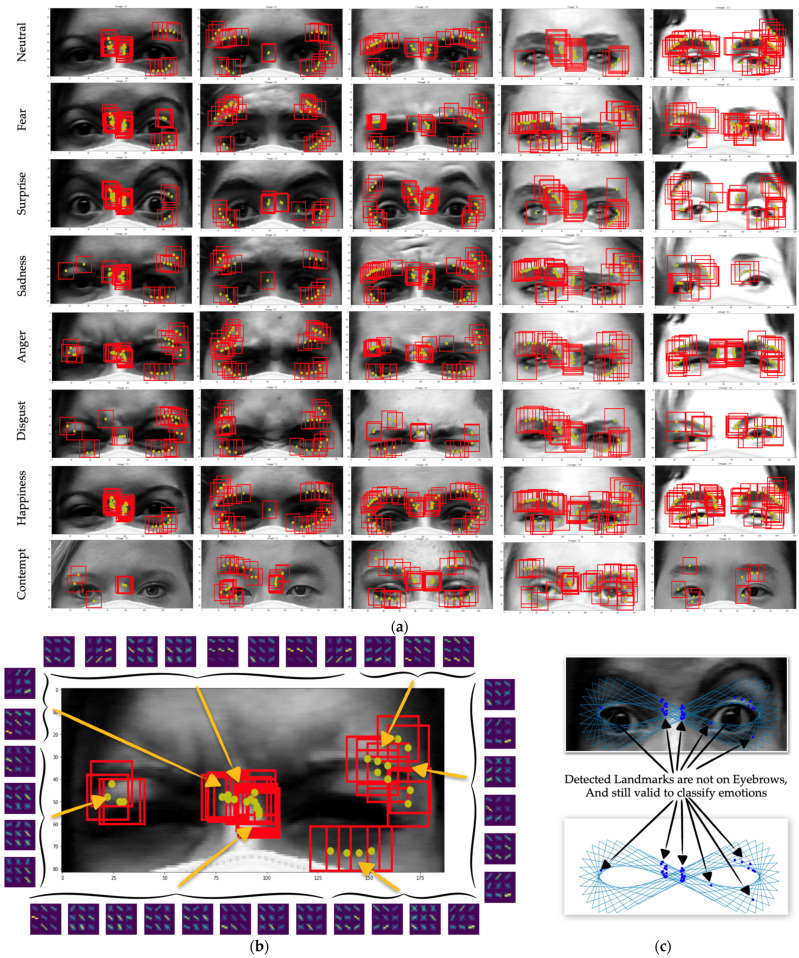
All of the detected landmarks for eight emotions in the CK+ data set; (**a**) images of upper faces with the infinity shape and detected landmarks; (**b**) detected landmarks; (**c**) detected landmarks that are not on the eyebrows.

**Table 1 sensors-22-04633-t001:** Comparison of input features and type of methods.

Methods	Input Features	Type	Limitation
Light-CNN [19]	Lower + Upper	Face-based	Mainly rely on Lower
eXnet [15]	Lower + Upper	Face-based	Mainly rely on Lower
Pre-trained CNN [19]	Lower + Upper	Face-based	Mainly rely on Lower
PG-CNN [14]	Lower + Upper	Face-based	Mainly rely on Lower
DLP-CNN [12]	Lower + Upper	Face-based	Mainly rely on Lower
gACNN [13]	Lower + Upper	Face-based	Mainly rely on Lower
RASnet [11]	Lower + Upper	Face-based	Mainly rely on Lower
DeRL [16]	Lower + Upper	Face-based	Mainly rely on Lower
ResiDen [20]	Lower + Upper	Face-based	Mainly rely on Lower
SHCNN [21]	Lower + Upper	Face-based	Mainly rely on Lower
ResNet-PL [22]	Lower + Upper	Face-based	Mainly rely on Lower
RAN [23]	Lower + Upper	Face-based	Mainly rely on Lower
SCN [24]	Lower + Upper	Face-based	Mainly rely on Lower
DACL [25]	Lower + Upper	Face-based	Mainly rely on Lower
ARM [26]	Lower + Upper	Face-based	Mainly rely on Lower
RTFER [11]	Lower + Upper	Constituent-based	Mainly rely on Lower
DenseNet [27]	Lower + Upper	Constituent-based	Mainly rely on Lower
OADN [28]	Lower + Upper	Constituent-based	Mainly rely on Lower
Proposed Method	Upper	Constituent-based	Rely only on Upper

**Table 2 sensors-22-04633-t002:** State-of-the-art results of the CK+ and RAF-DB data sets.

CK+			RAF-DB		
Methods	Accuracy (%)	Parameters (M)	Methods	Accuracy (%)	Parameters (M)
Light-CNN [19]	92.86	1.1	ResiDen [20]	76.54	12.1
eXnet [15]	95.81	4.6	Light-CNN [19]	77.23	1.1
Pre-trained CNN [19]	95.29	7.1	SHCNN [21]	81.17	8.7
PG-CNN [14]	97.03	20.5	DenseNet [27]	81.93	7
DLP-CNN [12]	95.73	17.5	gACNN [13]	85.07	21.4
gACNN [13]	96.40	21.4	eXnet [15]	85.59	4.6
RASnet [11]	96.28	10.4	ResNet-PL [22]	86.97	13.2
DeRL [16]	97.30	21.1	RAN [23]	86.99	15.6
RTFER [11]	93.85	0.5	SCN [24]	87.03	16.7
Proposed Method	99.30	1.3	OADN [28]	87.16	17.8
			DACL [25]	87.78	16.6
			ARM [26]	88.23	16.3
			Proposed Method	95.58	1.3

**Table 3 sensors-22-04633-t003:** Accuracy metrics of the proposed method on the CK+ and RAF-DB data sets.

		CK+					RAF-DB		
Emotions	Precision	Recall	F1-Score	Accuracy	Emotions	Precision	Recall	F1-Score	Accuracy
Neutral	0.996	0.997	0.997	0.997	Neutral	0.968	0.967	0.968	0.967
Anger	1.00	0.996	0.998	0.996	Anger	0.932	0.941	0.937	0.941
Contempt	0.958	0.958	0.958	0.958	Disgust	0.890	0.952	0.920	0.952
Disgust	0.969	0.998	0.983	0.998	Fear	0.972	0.911	0.941	0.911
Fear	0.955	0.984	0.969	0.984	Happiness	0.968	0.967	0.968	0.967
Happiness	0.993	0.976	0.984	0.976	Sadness	0.955	0.951	0.953	0.951
Sadness	0.972	0.994	0.983	0.994	Surprise	0.947	0.933	0.940	0.933
Surprise	0.997	0.981	0.989	0.981					

## Data Availability

This study utilizes the publicly available datasets from CK+: https://www.kaggle.com/c/visum-facial-expression-analysis (accessed on 21 December 2021), and RAF-DB: http://www.whdeng.cn/raf/model1.html (accessed on 3 April 2022).

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
