# Peer review of "Emotion Recognition for Partial Faces Using a Feature Vector Technique"

_sensors, 2022, doi:10.3390/s22124633_

Round 1

Reviewer 1 Report

1.     It could be better if add the parameter about the sample’s age, makeup of women et .al

2.     It could be better if this paper can give a deeply comment about the effect of the dimension of face to the recognize accuracy

Author Response

We sincerely appreciate your valuable time and effort in reviewing this work. We strongly believe that your comments and suggestions on this work will effectively improve the study results. Thank you for highlighting the flaws that we had overlooked. We did our best to include all of the reviewer's valuable advises and comments in our work. We would like to describe what we have done with the manuscript to make sure that we follow all of the reviewers’ comments as followed. Again, we sincerely appreciate your valuable comments. Thank you. For the detail, please kindly find them in the attachment file.

Reviewer 2 Report

This paper presents an emotion recognition network for partial facial images. The proposed method, which includes a flexible landmark detector and HOG to extract features for emotion classification, focused on the upper part of the face. To demonstrate the effectiveness of the proposed method, the authors examine the performance of two datasets such as CK+ and RAF-DB.

The proposed method seems straightforward and less complicated. However, I think the analysis and experiment used for the proposed method using only the upper part of the face are not convincing. It would be better to supplement the questions below.

-   In [Section 2.1], the author's reported results of 68 landmark analyses of facial images are not convincing at all. According to the author, the lower part of the face in the CK+ dataset has a high density and distribution of 68 landmarks. However, there is no quantitative evidence to support this hypothesis. Even the author presents an analysis based just on a single person's landmark. It would be better for the author to compute and reanalyze the results of multiple landmarks and their distribution and density.

-      In [Section 2.2.5], it is difficult to understand the output of the proposed landmark detector. Also, it would be helpful to describe the L1, L2, and P symbols in detail.

-     In [Section 3], the experiment was not clearly described. It is difficult to determine if the proposed model is trained using the synthetic masked face image dataset. In addition, it is unknown if the other approaches shown in Table 2 are trained using the synthetic masked face image dataset. The author needs to clearly write whether the synthesized data is used for training.

-   In line 434, the author said, “HOG features of the landmarks are the significant assets that should be used to classify emotions.” However, it is difficult to determine from experiment results whether HOG features are effective. It would be better to verify this through further experiments on HOG.

[Minor issue]

-       I think it would be helpful to improve the clarity of the paper's illustrations.

-       In line 84, the author said that the remaining chapters explain the Related Work, yet it appears there is no Related Work.

-       In Fig.2, there is no image of the “contempt”.

-  In line 309, it seems necessary to write a specific smoothing method to solve the problem of horizontal projection.

-   The authors should revise the entire format of the manuscript to make it sound more formal and concise. Also, the reference style should be consistent, therefore, this needs to be fixed as well. For example, there is no spacing between the words in [25].

Author Response

We sincerely appreciate your valuable time and effort in reviewing this work. We strongly believe that your comments and suggestions on this work will effectively improve the study results. Thank you for highlighting the flaws that we had overlooked. We did our best to include all of the reviewer's valuable advises and comments in our work. We would like to describe about what we have done with the manuscript to make sure that we follow all of reviewers’ comments as followed. Again, we sincerely appreciate your valuable comments. Thank you. For the detail, please kindly find them in the attachment file.

Reviewer 3 Report

  1. The title is okay, but it needs to mention specific emotions such as eyes
  2. Despite the fact that the authors explain the abstract very well, they need to explain why they chose CK+ and RAF-DB as their databases.
  3. During the introduction of the paper, the authors proposed landmark detection, but I did not see it in the abstract
  4. I believe the methodology is fine
  5. However, they should specify that it is for the eyes only
  6. For table 3, it may be more useful to provide statistical representation of both CK+ and RAF-DB
  7. The conclusions in the paper can be verified 
  8. The references are fine
  9. no more comments

Author Response

(The authors gave the same response as above.)

Round 2

Reviewer 2 Report

The author solved our doubts through the revision, and it was well reflected in the paper.